# On the Implementation of a Blockchain-Assisted Academic Council Electronic Vote System

**João Alves** [1,*,†]**, António Pinto** [1,2,*,†]

1 ESTG, Instituto Politécnico do Porto, 4610-156 Felgueiras, Portugal
2 CRACS & INESC TEC, 4200-465 Porto, Portugal
* Correspondence: 8000055@estg.ipp.pt (J.A.); apinto@inesctec.pt (A.P.)
† These authors contributed equally to this work.

**Abstract:** The digitisation of administrative tasks and processes is a reality nowadays, translating into added value such as agility in process management, or simplified access to stored data. The digitisation of processes of decision-making in collegiate bodies, such as Academic Councils, is not yet a common reality. Voting acts are still carried out in person, or at most in online meetings, without having a real confirmation of the vote of each element. This is particularly complex to achieve in remote meeting scenarios, where connection breaks or interruptions of audio or video streams may exist. A new digital platform was already previously proposed. It considered decision-making, by voting in Academic Councils, to be supported by a system that guarantees the integrity of the decisions taken, even when meeting online. Our previous work mainly considered the overall design. In this work, we bettered the design and specification of our previous proposal and describe the implemented prototype, and validate and discuss the obtained results.

**Keywords:** academic councils; Electronic Voting System; digitisation; blockchain





## 1. Introduction

In Portugal, a common organisation structure present in higher education institutions is the council, examples being the scientific council or the pedagogical council. A significant part of the activities within the Faculties of Portuguese Universities and Polytechnic Institutes is governed by these academic councils. Usually, each University comprises multiple Faculties, each one having multiple academic councils. Their decision-making process is based on majority decisions by nominal member voting without anonymity. Each member manifests his vote by a show of hands. They operate in a way much similar to the nominal group technique first developed in 1971 [1]. The councils are usually composed of members, a president and a secretary. The president will present one or more topics to be discussed and voted on, each member will manifest himself either for or against the decision, and the votes are counted. Decisions are then written in a minutes book that accounts for the votes of all members on all issues but without a formal assurance that the vote of each member was considered or rightly considered. Moreover, if no additional measures of integrity are considered, a section of a minutes book may, in principle, be altered without its members being aware of such change. Such alteration, if performed by someone with bad intentions, may even occur long after the end of the mandates of the related members.

Electronic Voting Systems (EVS) are a way to perform a more effective act of voting in democratic societies [2]. They reduce the cost of electoral acts by requiring fewer human resources per act. While EVS have yet to become the de facto way of voting in modern days, the forced push to remote working due to the COVID pandemic may increase the pace of its adoption [3]. That leads the authors to consider a blockchain-assisted EVS to support the decision-making process of academic councils that are gathering remotely or online.

In [4], the same authors already analysed the general adequacy of blockchain technologies when applied to EVS. The conclusion was that EVS can benefit from the integration of blockchain but requires a case-by-case analysis. No solution that fulfils the common requirements of EVS, plus voter anonymity, while being scalable and without requiring the spending of digital currency, was identified. Key shortcomings were related to the anonymity requirement that, while not present in academic council voting, is a common requirement of general EVS.

The adopted reference scenario, depicted in Figure 1, comprises an EVS for academic councils based on a web platform, made available to the members, where it is possible to access a topic under discussion and to record a member's vote. The President, or Secretary, of the academic council starts a meeting and puts topics up for discussion, one at a time. Each topic, after being discussed, is then subjected to a nominal vote by each member. Members cast their votes on their devices when the President closes each topic. The votes, requiring only confidentiality and not anonymity, are then collected and counted, and a decision is made. After the decision is computed, it is also stored in a ledger or blockchain. The decision will consist of the result of a secure keyed hash function over all collected votes.

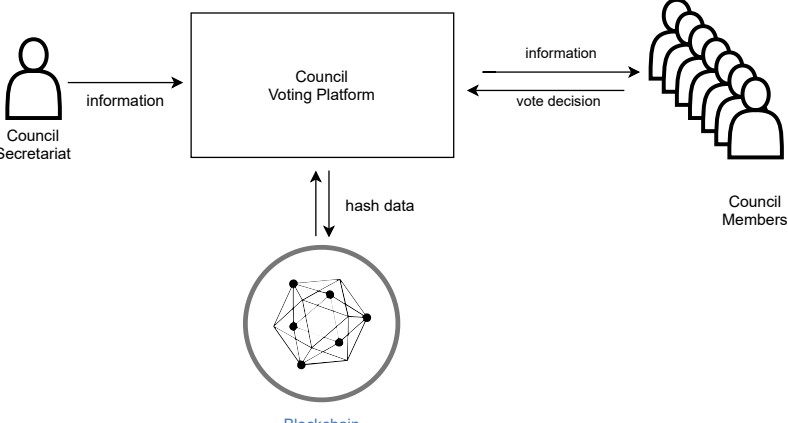

**Figure 1.** Reference Scenario.

The same authors, in [5], proposed and designed a blockchain-assisted system to support the decision-making of academic councils that operate by nominal voting in assemblies, gathering remotely or online. In the work herein, we:

- Better designed and specified the previous proposal;
- Described the implementation of a proof of concept prototype;
- Tested and validated the proposed solution using the implemented prototype;
- Performed a security analysis of the proposed solution.

This paper is organised into sections. Section 2 describes EVS and presents its main principles. Section 3 describes and compares existing solutions that are similar to ours or that implement some of the features also included in the proposed solution. In Section 4 we enumerate the identified requirements and present the proposed solution, including its architecture and its security analysis. Section 5 describes the implemented prototype. Section 6 discusses the obtained results. Section 7 concludes our work.

## 2. Background

A Blockchain can be seen as a ledger that is decentralised, transparent and tamper-proof [6]. The term Blockchain was used by Nakamoto in 2008 to describe a data structure that comprises information regarding transactions, organized in blocks, and by storing the secure hash of each block into the information included in the following block, promotes data integrity and creates a chain of blocks [7]. Since then, Blockchain has gained public attention and is frequently referred to as the underlying technology supporting the bitcoin

cryptocurrency. Nakamoto's solution solved the double-spend problem of digital currencies by relying on a distributed and open network of equal peers [8]. A peer can assume the role of a miner, the one responsible for guaranteeing the security of the overall system. The miners execute the Proof of Work (PoW) consensus algorithm to both validate the new blocks that are being added to the blockchain, and also to receive bitcoins for their work. Miners compete amongst themselves in order to be the first to produce a valid block, adding them to the chain and receiving its reward. This decentralised operation creates multiple verification points for every transaction.

### 2.1. Blockchain

The bitcoin blockchain comprises the concepts of signed blocks and PoW [7]. In practice, the nonce value that is part of any block is only useful in order to achieve consensus among peers. In practice, a miner will create a block by adding a set of unregistered transactions and then setting the nonce value to one, for instance. It will then calculate the hash value of the block and if the resulting hash value comprises a predefined set of leading 0 (zeros), the block is considered signed. If not, the miner will then iterate the nonce value, and recalculate its hash until it achieves the desired leading number of zeros. The SHA-256 secure hash algorithm is the one used in bitcoin [9]. These functions made the system secure but very inefficient [10]. Specialised hardware, known as mining platforms, was specifically created to handle the work of processing higher numbers of transactions [11].

Figure 2 shows a simple representation of bitcoin's Blockchain. It comprises a small list of blocks of transactions linked together using the previous block's hash. The depicted Merkle root consists of a hash value of the transactions stored within each block and is used as a way of summarizing all transactions [12].

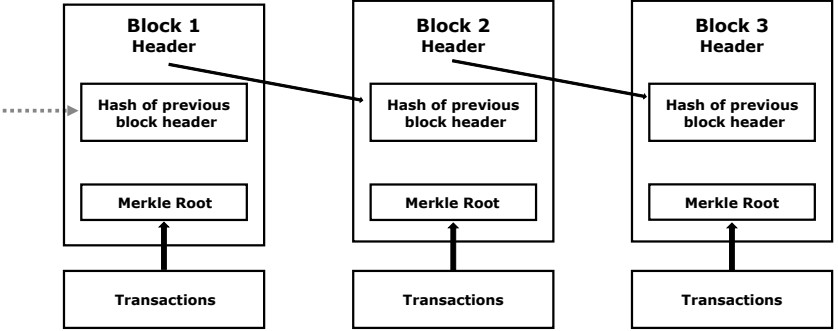

**Figure 2.** Representation of the bitcoin's Blockchain (adapted) [7].

As described earlier, new blocks are only added when mined, which results in redundant storage and in a trustless proof mechanism [13]. Third-party trust authorities, or other trust intermediaries, such as banks or brokers, are no longer required for the system to operate. Other key characteristics include its digital nature and being chronologically time-stamped, tamper-resistance, auditable and real-time update [14]. All peers, with no distinction, may have a copy of all information within the Blockchain. Blockchain is expected to be fully digitised, without any manual interaction. Information should be stored in the Blockchain in (near) real-time. The Blockchain repository stores information about each transaction with links to previous blocks in a chronological manner, making a trail of the underlying transaction sequence. All stored blocks must be cryptographically sealed, making it impossible to delete or alter blocks within a network, ensuring a high level of trust and robustness, and creating true digital assets. By being stored in a decentralised way, it becomes failure-resistant, even when a large number of peers fail, for as long as there are enough peers still operating. By being consensus-based, all transactions are only executed if all participants on the network unanimously approve it.

Blockchains can also be grouped into two types: permissionless and permissioned [15]. A permissionless Blockchain, as is the case of Bitcoin's Blockchain, consists of a distributed network of equal peers in terms of permissions to read and write data to the blockchain.

Permissioned Blockchains differentiate peer access permissions in terms of read and write access, which normally involve a consortium of multiple organisations, where block transitions are created and verified by authorised gatekeepers instead of anonymous miners. In essence, in permissioned Blockchains, some peers have privileges to change the information on the Blockchain while others can only read the information.

## 2.2. Smart Contracts

Blockchains can also include smart contracts [16]. In 1994 [17], Nick Szabo described the concept of smart contracts, several years before the publication of Nakamoto's bitcoin in 2008 [7]. Smart contracts are computer programs that can be consistently executed by a network of nodes, without the arbitration of a trusted authority. Smart contracts are pieces of computer software created to automate a self-enforcing contract. It means that it will trigger a certain action when predetermined conditions are met. An alternative definition for smart contracts was stated by Vigliotti:

> "A computerised transaction protocol that executes the terms of a contract. The general objectives of smart-contract design are to satisfy common contractual conditions (such as payment terms, liens, confidentiality, and even enforcement), minimise exceptions both malicious and accidental, and minimise the need for trusted intermediaries. Related economic goals include lowering fraud loss, arbitration and enforcement costs, and other transaction costs." in [18].

New concepts were introduced by new smart contract-based platforms, an example being the Ethereum tokens that follow the ERC-20 [19]. These tokens are often distributed through events named Initial Coin Offerings (ICO) and can represent points, money or reputation, among others. Its use can facilitate payment processing in Distributed Applications (DApps), decentralised markets or Decentralized Exchanges (DEX) [20].

Smart contracts are versatile [21] and assumed an important role in the Blockchain space [21] because of their resilience to tampering. Smart contracts are appealing in many scenarios, especially in those which require that transfers of money must respect certain predetermined rules. The equipment leasing industry [22], for example, has made use of smart contracts in real-world scenarios in order to make leasing arrangements more efficient [23]. In the healthcare sector, the technology is being explored as a measure against data manipulation in clinical trials [24]. Smart contracts can even be used to enforce intellectual property agreements [25,26] by establishing a definitive record of shared property rights. Other examples include the insurance sector [27], agriculture [28], and gaming [29].

## 2.3. Smart Contracts in Hyperledger Fabric

Hyperledger Fabric [30] is a permissioned blockchain platform supporting smart contracts. It can be categorized as a permissioned distributed ledger platform, designed to be used in enterprise contexts. Differentiating factors are its modular and configurable architecture, the support for authoring smart contracts using general-purpose programming languages such as Java or Go, and the support for pluggable consensus protocols. A key characteristic of Hyperledger Fabric is the fact that it does require the use of a cryptocurrency.

When compared to other platforms, such as Ethereum, Hyperledger Fabric obtains a better performance [31] and supports privacy by adopting a permissioned mode of operation. In particular, to do so, it uses a Byzantine Fault-Tolerant (BFT) algorithm, fine-grained access control, and public-key cryptography. Moreover, the modular architecture allows Hyperledger Fabric to be customized to any type of application [32].

Before organisations can interact with each other, they must define a common set of contracts (terms, data, rules, concepts, and processes). This group of contracts establishes the business model that will govern all future interactions. For example, a smart contract can ensure that a new car is delivered within a specified time frame or that funds revert back to the buyer, improving the flow of goods and capital. Hyperledger Fabric documentation refers to smart contracts and chaincode interchangeably but these are different concepts.

A smart contract defines the transaction logic of an object, which is then packaged into a chaincode. Multiple smart contracts can be defined in the same chaincode and when a chaincode is deployed, all smart contracts within it are made available to applications. A smart contract accesses two distinct parts of the ledger, the Blockchain and a world state, which maintains a cache of the current value of all states. Smart contracts basically "put", "get", and "delete" states into the world state and can also query the Blockchain for records of transactions. A "get" consists of a query to retrieve information about the current state of a business object. A "put" either creates a new business object or modifies an existing one. A "delete" requests the removal of a business object from the ledger's current state.

### 3. Electronic Voting Systems

Public administrations encourage the development of Electronic Voting Systems (EVS) to bring efficiency, reduce bureaucracy and costs, and enable voters with an online system capable of providing an electronic ballot that still maintains all required characteristics [33]. The adoption of EVS is a step that must be taken on the path to the dematerialisation of all processes. EVS must maintain some of the key characteristics of the traditional ballot systems. Vote secrecy appears to be a key characteristic in most cases.

According to Gritzalis (2002) the main constitutional requirements of secure EVS are: Generality, Freedom, Equality, Secrecy, Directness, and Democracy [2]. Generality means that every voter has the right to participate in an electoral process. Eligibility must be founded and regulated by law. Voting technology should be accessible to all voters and should be seen as an alternative way for an elector to cast his vote. The democratic principle, in which all eligible voters should be included in the electoral process, must ensure that public electronic infrastructures are available (public Internet kiosks or state infrastructure offices or others) so that they can exercise their right to vote. Eligibility can be assured by the registration of eligible voters and by their identification at the time of their registration. Registration and authentication are essential procedures so that the principle of universality is respected and that election acts cannot be manipulated. The purpose of maintaining voter registration is to ensure that only eligible voters can vote and that they cannot vote more than once [2].

Freedom means that the election process must ensure that it occurs without any violence, coercion, pressure or any other manipulative intervention that may be inflicted on the voter by a third party. EVS assume an important role here, guaranteeing the uncoercibility [34], the prevention of vote buying, and also the guarantee that no voter can be audited in order to perceive their tendency to vote [2].

Equality, in a general election context, represents the generic principle of equality and represents a pillar in democracies. Two main requirements must be ensured: equality for each party/candidate, and equal voting rights for each voter. The electronic registration of each voter card must have the same paper registration requirements. Electronic ballot papers must also be similar to those printed and on the same form accessible. Transparency must be supported so all parties should have the same opportunity and equal access to the elements of the voting procedure. The duration of the electoral act must be defined and the voting period must not be exceeded [2].

Secrecy means the inability of someone to discover the vote cast by a voter. Depending on each type of election, secrecy may or may not be required. It is necessary in the case of free political elections. Contrarily, in group decision-making by vote, secrecy may not be a requirement [2].

Directness, in traditional voting systems, means that there be no intermediaries in the voting process. This principle should also be adapted to an electronic voting procedure. The relevant requirement is that all online ballots are recorded and counted directly. A problem may arise if the voting period differs from the voting procedure (online or offline) used to vote. The results of online voting can influence the outcome of the entire electoral process and limit the integrity and legitimacy of the whole process. To avoid this, a system

can be developed allowing the recording and maintenance of ballots, while prohibiting any counting before the end of the voting period (offline) [2].

Democracy means that EVS must respect the requirements of a traditional electoral system. Voters should be able to understand how elections are conducted. Traditional voting procedures work transparently for both voters and other constituencies. On the contrary, electronic voting procedures are not transparent because the average voter does not have the knowledge to understand how the system works. Therefore, in electronic voting, voters deposit more trust in the used technology and the people involved (election officials, technology providers) [2].

However, additional requirements must also be met due to the nature of electronic voting; mainly because other features are being added and new requirements are being imposed [2]. The overall security of the voting process must be assured and fraud-proof. From the voter's point of view, EVS must confirm his participation in the electoral act. It should enable indiscriminate access to the voting platform, as well as maintain a record of votes. There must be a reliable certification procedure for both hardware and software. The entire infrastructure, as well as, any system functionality, must be registered and documented. All operations, such as authentication, recording of votes, and others, must be monitored, while still enforcing confidentiality. The infrastructure must be open for inspection by authorised bodies. Voters, parties and candidates must be assured that there has been no malpractice. Adequate system security must be ensured, be simple and easy to use. Participation in the voting process should be able to be confirmed. Uncoercibility should be ensured, particularly if remote voting is an option. The ability to consciously cast a non-valid vote should be provided for. No voter should be able to duplicate or change his vote or the vote of someone else (integrity). The voter should be able to verify that his vote is calculated in the final tally (verifiability). Voters should be able to have undiscriminating access to the voting infrastructure (accessibility). Registration, authentication and voting procedures should be evidently separated. Votes should be validated separately and independently from voter authentication. No intermediaries should be involved in the voting process, no person can be authorised to vote for another person. Each and every ballot should be recorded and counted correctly.

### 3.1. Blockchain-Based EVS Supporting Anonymity

Ayed proposed a Blockchain-based EVS [35] in 2017, where four main requirements for EVS were identified, these being: authentication, anonymity, accuracy and verifiability. Accuracy and verifiability can easily be extracted from the Blockchain. Authentication, to assure that only registered voters can cast votes, is a parallel process not supported by the Blockchain. The solution proposed by Ayed consists of hashing the voter's name, number, vote and the hash of the previous block. Of this information, only the vote is a priori unknown and, by being a limited set of options, it should be feasible to break as one would only need to run as many hash operations as the number of voting possibilities. Moreover, the anonymity of the voter's identity is said to be achieved by not allowing any links between voters and ballots. Bitcoin transactions are considered pseudo-anonymous as only wallet addresses are used in transactions and not the identifiers of the people behind each transaction. However, upon discovery of the owner of a wallet, anyone can view all transactions issued by that wallet address.

To introduce anonymity in bitcoin, solutions such as Zerocoin were proposed by Miers et al. [36]. Zerocoin is an extension of the bitcoin protocol, which enables anonymous currency transactions. These services are also known as bitcoin laundry services [37].

Takabatake et al. in [38] proposed the use of Zerocoin as a basis EVS and, by doing so, the transparency, integrity and anonymity requirements are resolved. Transparency and integrity are achieved due to the distributed and public nature of the Blockchain. The anonymity is solved with the use of Zerocoin. This solution, however, is not completely anonymous because the IP address of public nodes can be known. Moreover, it is not clear if registration, authentication and voting procedures are separated.

BlockVotes, proposed by Wu in [39], is yet another example of Blockchain-based EVS. BlockVotes uses a ring signature algorithm to generate values that are stored in the Blockchain. While satisfying requirements such as ballot privacy, anonymity individual verifiability, eligibility completeness, uniqueness, robustness, and coercion-resistant, it does not satisfy fairness nor receipt-freeness requirements.

Jason and Yuichi proposed another EVS based on Bitcoin [40]. It consists in using the Bitcoin protocol in conjunction with a blind signature scheme [41]. Blind signatures make a third party capable of attesting to the contents of some data (a vote) without having access to the data. These solutions enable anonymity in voting but require the usage of cryptocurrency to cast a vote. The authors suggest the use of prepaid bitcoin cards and proposed three distinct entities: the voter, an administrator and a counter. The administrator is the entity that makes use of blind signatures.

Tasarov et al. proposed another solution for Blockchain-based EVS [42] that, instead of bitcoin, uses Zcash [43] as the underlying payment system. The proposed solution achieves transaction anonymity using Zcash, unmodified. This anonymity relied on zero-knowledge proofs as a substitution for the PoW scheme present in bitcoin. It supports both anonymous and transparent transactions and has two types of addresses, which differs from the bitcoins single address and uses four distinct steps: registration, notification, voting ad counting. For voter identification and verification, it uses X.509 certificates.

Multiple others proposals and reviews of EVS [44] can be found in literature, namely those described in [35,37,39,42,43]; all supporting voter anonymity. Moreover, in [45], the authors review various voting protocols and methodologies based on Blockchain. Most of them provide features such as non-re-usability, verifiability, self-tallying, eligibility and principally anonymity, which is present in most of them.

*3.2. Related Work*

Vote anonymity, which is related to not being able to identify the one casting a vote, is not a requirement for all types of elections. The adopted reference scenario does not require vote anonymity and, therefore, none of these solutions can be directly used in the reference scenario. Other more suited solutions exist and were classified as related work, these will be analysed next.

Verify-Your-Vote (VYV), described in [46], is presented as a secure online blockchain-based e-voting protocol. The authors use the blockchain as a bulletin board, supporting election acts that are verifiable. Voters can only vote after being authenticated. Requirements such as vote privacy, receipt-freeness, fairness, and individual and universal verifiability are also supported. Authors identified, as the main goal, and as a requirement, that their solution must be able to be formally verified at the protocol level, using a ProVerif tool.

Hardwick et al., in [47], proposed another EVS based on blockchain technology that addresses some requirements that are less frequent. One such requirement is the capability of a voter to change his vote after being cast. Additionally, the authors strive for a maximum degree of decentralisation where the voters control the network by being part of it, as its peers.

Bistarelli, et al. proposed an EVS based on Bitcoin [48]. It comprises three phases: pre-voting, voting and post-voting. The resulting implementation is completely decentralised as you can vote without any intermediaries. All votes can be verified by anyone who reads the public ledger. This solution can only be applied when voter anonymity is not a requirement. Uncoercibility, confidentiality and neutrality are also not satisfied. Once a vote is issued, it is broadcasted to the peer-to-peer network and, by not being confidential, it can influence future voters.

The Secure Electronic Voting System (SecEVS), proposed in [49], is focused on secure e-voting in university campus elections. They make use of Merkle root hashes, such as Bitcoin, guaranteeing the privacy of the transmitted data, voter confidentiality, and the uniqueness of each vote, avoiding duplication. The scenario of SecEVS appears similar

to the one proposed herein, but the key difference resides in the fact that the envisioned reference scenario is not compatible with voter anonymity.

Table 1 compares the related work with respect to multiple characteristics. Namely confirmation, eligibility, verifiability, unreusability, uncoercibility, integrity, uniqueness, validation, and ballot counting. Confirmation requires a process to confirm participation in the voting process. Eligibility requires a process to verify if a voter is eligible to vote. Verifiability requires processes to allow each voter to verify his own vote. Unreusability requires that each voter can only vote once, preventing future changes in votes already made. Uncoercibility means that the system must prevent any attempt to coerce the vote of others. Integrity refers to maintaining each vote as it was cast, making it difficult or impossible to tamper with. Uniqueness requires the guarantee that a voter can only cast one vote, per each voting act. Validation requires a process that enables the verification of the overall voting process. Ballot counting requires a vote-counting process that can be validated.

**Table 1.** Comparison of related work.

| Related EVS | VYV | Hardwick | Bistarelli | SecEVS |
|---|---|---|---|---|
| Confirmation | Y | Y | Y | N |
| Eligibility | Y | Y | Y | Y |
| Verifiability | Y | N | Y | Y |
| Unreusability | – | N | Y | Y |
| Uncoercibility | N | Y | N | N |
| Integrity | Y | Y | N | Y |
| Uniqueness | Y | Y | N | Y |
| Validation | Y | Y | Y | Y |
| Ballot counting | Y | – | N | Y |

Requirements such as the confirmation of participation in the voting process, only allowing eligible voters to vote, permitting voters to verify that their vote is considered in the result, or that votes are not reused are almost satisfied by all solutions. For instance, the solution proposed by Bistarelli [48] supports vote confirmation, voter eligibility, unreusability, integrity, separation, and validation and requires the use of currency to cast votes. Uniqueness, ballot counting, and uncoercibility are not satisfied by this solution. The SecEVS solution considers a reference scenario that is closer to the one adopted in our work, in the sense that both address voting in academic institutions and do not require the expenditure of digital currency for its operation. Nonetheless, they differ because SecEVS is geared towards a large electoral vote, while, in our case, multiple voting processes are to be expected per plenary meeting. Moreover, voter anonymity support provided by SecEVS collides with the requirements of the adopted reference scenario.

The analysis of the existing literature and related work, complementary to previous work on this subject in [4], leads the authors to conclude that the proposed solution is novel and not previously addressed by our peers.

## 4. Blockchain-Assisted Academic Council Electronic Voting System

The proposed solution, named the blockchain-assisted Academic Council Electronic Voting System (bACEVS), aims to specifically address the recording of decisions made in academic councils. With the proposed solution, a faithful record of voting acts is maintained as well as a significant increase in usability and verifiability, of the entire decision-making process, is expected. Please recall that, while voter anonymity and voting privacy are not required within the envisioned reference scenario, characteristics such as voter eligibility, ballot counting, integrity, verifiability and uniqueness of votes still need to be addressed. Such considerations lead to the identification of the following list of key requirements:

R1 Support multiple Academic Councils, which belong to multiple faculties, within one institution—With our solution, it is possible to register several academic councils,

such as scientific or pedagogical councils, which will then have terms of office with a well-defined duration as the members who will be part of that term.

R2 Support an operation based on the fact that members are elected for a mandate (period of time)—The system should allow choosing which members belonged to a mandate, what type of member it will be, President, Secretary or normal member. Each mandate will have a well-defined period of activity.

R3 Support the management of Academic Councils and their members—The system must implement CRUD operations for all tables allowing the entire management process of relevant information.

R4 Schedule meetings and record the list of members participating in each one—One of the important requirements is to be able to create Meetings, schedule them, invite the members who will participate in each meeting and register your participation in each one of them.

R5 Insert, modify and remove topics subject to be decided by a voting act—Possibility to have one or several topics to be discussed in each meeting and each one of them having the possibility to be voted on. Here the vote can be a yes or a no according to the position taken by each member or the issuance of a written opinion regarding that topic.

R6 Support the association of external digital documents that are relevant to the topics to be voted on—One important feature of our system is the total digitization of all documents related to electoral acts. Thus, we will have to provide the system with the possibility of associating digitalized physical documents either to each topic that will be voted on or to each meeting as a whole, as in this area it is very common to have physical documents from various sources.

R7 Register the participation of all members in a voting act, including their vote and the overall decision—The participation of each member is critical to the success of each voting process, thus, the record of the participation of each member shall exist as well as the position taken by each one in each voting process.

R8 Guaranty the immutability of records pertaining to voting acts—As the main objective, the guarantee that the final result of a voting process that took place in a given meeting cannot be changed needs to be ensured. Thus, the use of Blockchain with its characteristic of immutability will be fundamental for the success of this project.

### 4.1. Use Cases

The crossed analysis of the reference scenario (see Figure 1) with these key requirements resulted in the definition of a set of use cases. The main use cases that were identified are depicted in Figure 3. These comprise two actors: President and Member. The Member will be able to access an ongoing meeting, open the topic under discussion and then cast a vote when the topic is open for voting. Members will also be able to view the information recorded on the Blockchain regarding their past votes. The President includes all functions available to the members, plus meeting and topic management. The President can start a meeting, close a meeting and view information about current and previous meetings. The President, at the topic level, can open a topic for discussion within a meeting, collect its votes and generate the decision result that will imply the topic being closed and for its decision to be sent to the Blockchain, as soon as the President closes the meeting. Here, the President is also allowed to view information about decisions on past topics and the status of the currently open topic.

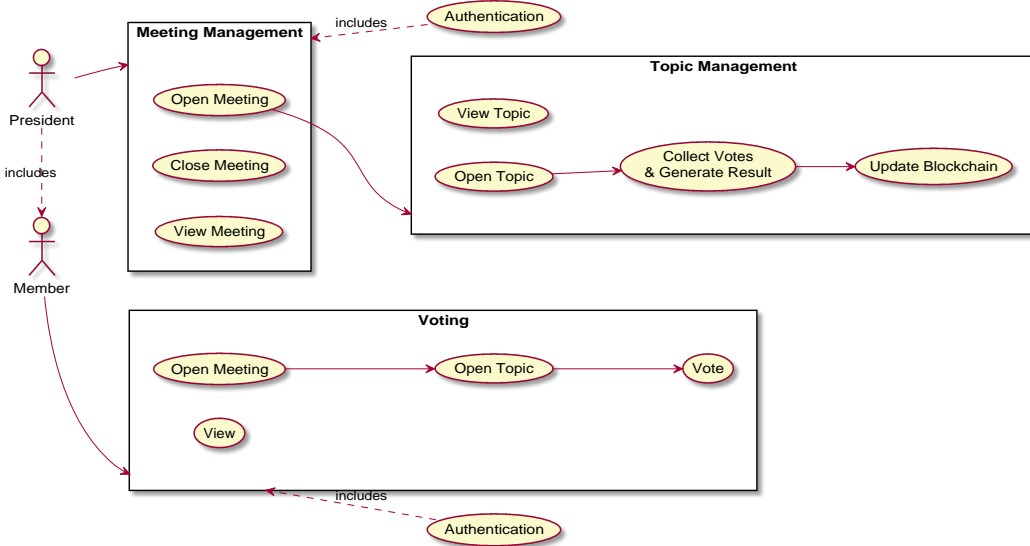

**Figure 3.** Use Cases.

## 4.2. Proposed Architecture

The architecture of the proposed solution is depicted in Figure 4. It comprises six components, these being: the Members, the President, a User Directory, the bACEVS, a Database, and the Blockchain. A Member represents a real person of the Academic Council that is eligible to participate in meetings by voting or issuing opinions on a presented topic. The President also represents a real person but one with a specific role within an Academic Council. The President is eligible to participate in meetings, voting and issuing opinions on topics. Additionally, this is the entity responsible for the coordination of meetings, performing actions such as opening meetings, presenting topics, closing topics, gathering votes or opinions issued, and closing meetings.

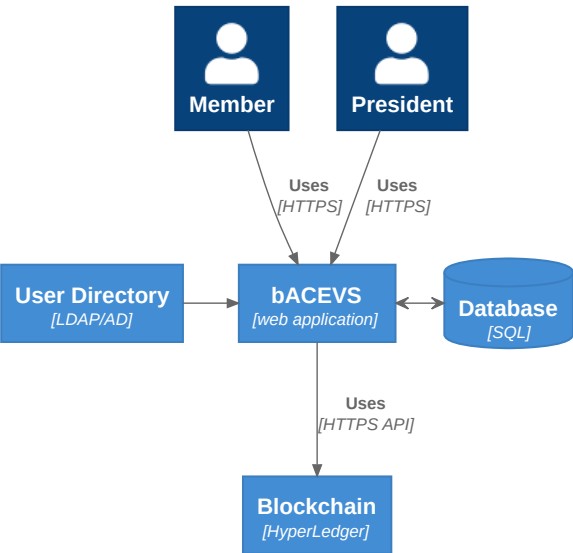

**Figure 4.** Architecture of the proposed solution.

The bACEVS system consists of a web-based application used to interact with all members of Academic Councils, supporting all relevant functionalities. It can be seen as the front-end application from the member's perspective, and as the gateway to the database of the Institution, to the user's directory database and also includes an API to the Blockchain. The web-based application is expected to be accessed by members with their smartphones, tablets or personal computers, using TLS-encrypted web connections.

The Blockchain API is responsible for the interface between the Hyperledger Fabric and the bACEVS. The User Directory consists of a database of users. It will be used by the proposed solution as a location for user-related information, including authentication-related information, such as usernames and passwords. This component is expected to already exist in the organisations either as an LDAP or as an AD database where bACEVS will authenticate members. The Database consists of a common relational database with SQL support. It will be used to store information about meetings, topics, the composition of the Academic Councils, member mandates and decisions taken in meetings. The decisions will be structured in JSON format. Ultimately, the Blockchain will be used to store all decisions made within the meetings of the Academic Councils. A decision representation will be stored in an immutable, verifiable and secure way, within the selected Blockchain, the Hyperledger Fabric.

### 4.3. Adopted Notation

Table 2 presents the adopted notation. Letters P and M represent system users, P for president and M for normal members. $ID_m$ represents the unique identifier of member $m$, for instance, the 4th member would be represented with $ID_4$ and could have a value of "088f0fd4-e8c5-4a0e-b042-2517b8e95971". $NM_i$ represents a number to be used only once (nonce) securely generated for the meeting $i$, for instance, the 2nd meeting nonce would be represented with $NM_2$ and could have a value of "55919b9-44eb-4771-853c-34cd95c697a3". $NT_i$ represents a nonce securely generated for the topic $i$ and can similarly be represented as the meeting nonce. $Ts_i$ represents a timestamp, for instance, the first timestamp would be represented with $Ts_1$ and contains the date and time in epoch format. $SEK$ represents a session encryption key created using a secure random number generator. $PriK_m$ represents the private key of member $m$. $PubK_m$ represents the public key of member $m$. $D_{T,M}$ represents the decision generated from the discussion of topic $T$ in meeting $M$. Details on the decision token can be found in the next section. $HMAC(D_{T,M}, SEK)$ represents the result of a secure Hash-based Message Authentication Code (HMAC) function, such as HMAC-SHA256, having $D_{T,M}$ and the $SEK$ as inputs.

**Table 2.** Adopted notation.

| P, M | Members |
|---|---|
| $ID_m$ | ID of member $m$ |
| $NM_i$ | Nonce for meeting $i$ |
| $NT_i$ | Nonce for topic $i$ |
| $Ts_i$ | Timestamp $i$ |
| $SEK$ | Session Encryption Key |
| $PriK_m$ | Private key of member $m$ |
| $PubK_m$ | Public key of member $m$ |
| $D_{T,M}$ | Decision on the topic $T$ of meeting $M$ |
| $HMAC(D_{T,M}, SEK)$ | HMAC of decision $D_{T,M}$ and key SEK |

### 4.4. Diagrams and Specification

A meeting starts when it is opened by the President or Secretary. In this step, a timestamp, a *nonce*, and a meeting symmetric encryption key are securely generated. The nonce consists of a random number that can only be used once, and also serves as an identifier for the meeting actions. After, the topics will be discussed and voted on, making them available to all members. Since a session can be interrupted or suspended after it has been opened, the recording of the decisions is taken topic by topic. This way, meetings can be resumed on a later date, without having members vote on all topics again, and just continuing the session on the topic that was suspended.

A notification message is sent to all members who participate in the meeting, alerting them that the meeting is open following the presentation and opening of each topic to be voted on, by the President. At this point, a new timestamped nonce is generated to identify each topic action and the topic discussion starts with the presentation and sharing of documents by the bACEVS, to all voting members. When a topic discussion ends, the President starts a vote collection action in bACEVS, triggering the sending of topic vote request messages to all participants. bACEVS then collects the votes cast by each voter and generates a JSON file, named decision record, with the votes of all participants. The decision record is then sent to the Database to be stored and is used as input to calculate the decision token to be sent to the blockchain for storage. After the successful end of a topic voting process, the President closes the topic and continues to the next topic, reiterating all steps until no more topics are available to be voted on. Finally, after the voting of all topics in the meeting, the President closes the meeting on the bACEVS, triggering a notification message informing all that the meeting is over.

Figure 5 describes the communications exchanged in the decision-making process between all entities. This exchange assumes that all members are online and authenticated in the bACEVS. The first step is the opening of a meeting, performed by the council's President (or Secretary), and consists of selecting an option in the bACEVS web-based user interface. In Step 2, the bACEVS will securely generate both a session key (SEK) and a nonce for the meeting ($NM_i$). These values, in Step 3, will be transmitted to all members actively participating in the meeting. Steps 4 up to 9 are repeated per topic that is included in the meeting. In Step 4, the President opens a meeting topic for discussion, by using the web-based user interface. The bACEVS system will securely generate a topic nonce, in Step 5, and communicate it to all active members, in Step 6. Next, the bACEVS will grant access to the documentation related to the topic being discussed to all active members. A period of discussion on the topic is expected at this time and, after this discussion, the President can request a vote collection (Step 7) which will trigger the bACEVS to issue vote request messages (Step 8) to all active members. In reply, all active members will return their vote (Step 9) to the bACEVS. All votes will be joined in a list. The list of votes and the current SEK will be the inputs to a secure HMAC function that, at last, will return the overall decision on topic $T$ of meeting $M$. The decision $D_{T,M}$ is the result of an HMAC function having as input a session key ($SEK$) and a list of the votes ($[V_1, V_2, \ldots, V_n]$, with $n$ being the number of members participating on the decision) of all members on a specific topic ($T$) of a specific meeting ($M$). Each one of these votes consists of multiple concatenated fields, all encrypted with the member's private key and structured as follows: $V_i = \{V : T : M : Topic\ Subject : NM_i : NT_i : T_s\}_{PriK_i}$ with $V$ being the vote cast by the member $i$ on topic $T$ of meeting $M$. $V$ can either be 1 (yea), 2 (nay) or 3 (abstention). Moreover, the votes being cast by the members are first sent to the BACEVS, meaning that no direct communication is expected between the members and the Blockchain.

Whenever a topic is closed, a decision record is also created and stored in the database. This decision record is a JSON-formatted data structure. Listing 1 shows a sample decision data structure with positive decisions. It includes the topic identifier (or nonce) and the identifiers of the members who voted for, against or abstained.

### 4.5. Security Considerations

While data availability and fault tolerance are naturally addressed in the blockchain, the proposed solution also makes use of a relational database to store the required remaining data. Moreover, the proposed solution assumes the presence of a node of the blockchain, and an installation of the bACEVS, per faculty within each institution. Data replications situations must then be addressed for the databases that are included in the bACEVS to promote fault tolerance, backup and, possibly, load balancing. The proposed approach is to adopt a database system that supports live synchronization between a distributed set of database nodes, which is the case of MySQL with the use of Group Replication, whenever there are at least three nodes.

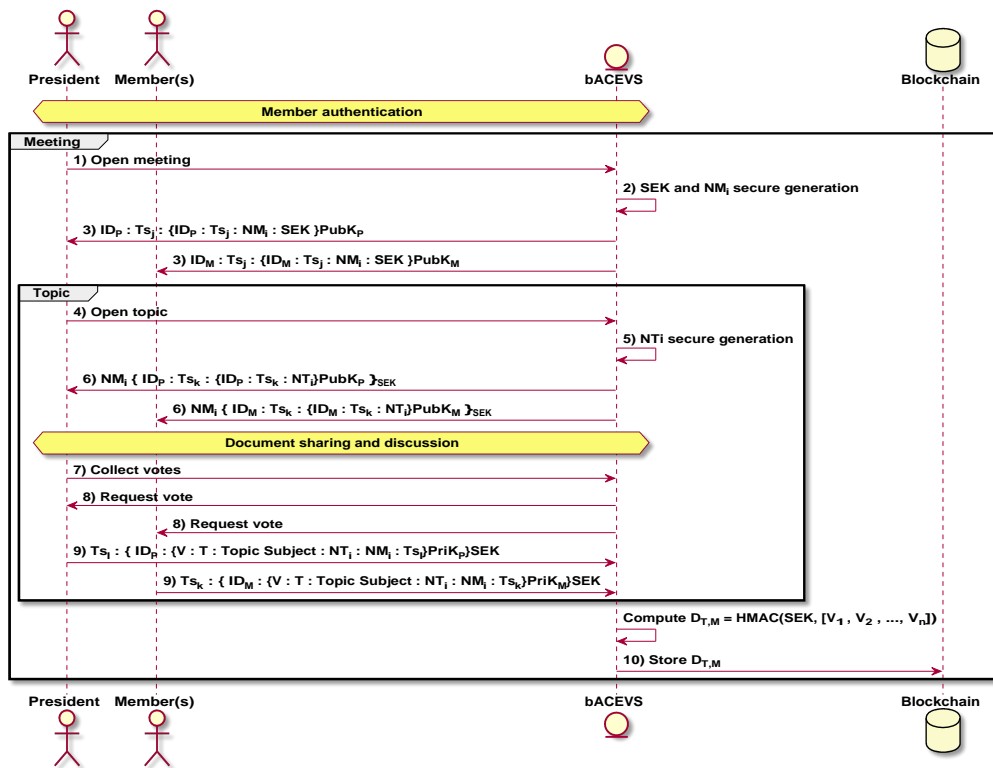

**Figure 5.** Decision token (D*i*) generation.

**Listing 1**: Sample decision record.

```
"topicID": "055919b9-44eb-4771-853c-34cd95c697a3",
"positive": {
  "members": [
    ["088f0fd4-e8c5-4a0e-b042-2517b8e95971"],
    ["bdcb984e-7fe8-4be5-bc19-79ee97368331"],
    ...
  ]},
"negative": {
  "members": [
    ["36989b61-6906-438d-8d32-15942fee92b0"],
    ...
  ]},
"abstention": {
  "members": [
    ["477698a6-2f31-11ec-8d3d-0242ac130003"],
    ...
  ]}
}
```

For the login process, the proposed solution will integrate existing User Directory services, such as an AD infrastructure, meaning that no user passwords will be stored or managed within the proposed solution. The application will also assure that votes can only occur whenever a session has been opened, checking all meeting-related information. Whenever a meeting is closed, the system will not accept more votes to be cast.

The session integrity resides in the use of freshly generated nonce numbers, which are not repeated. Nonces are generated by the system when sessions are opened, and then securely distributed to all Members. The nonce distribution message is encrypted with the public key of the destination member, meaning that no other person can have access to the information. The nonce will be used to sign each vote, satisfying this requirement. The anonymity of the decisions made by members is not required internally, but externally,

between distinct councils and distinct faculties. External confidentiality is achieved by the use of the channel concept of Hyperledger Fabric.

The collection of votes per topic within a meeting is triggered by the bACEVS system (step 8 of Figure 4). At the end of the vote collection, votes from all members that participate in the meeting are expected. If this is not the case, the bACEVS will issue a warning to the President. The President will then decide to either compute a decision based on the existing votes or to contact the Members in question in order to decide if a new vote request should be sent to those specific Members. In order to avoid duplicate votes, the bACEVS will only accept the first valid vote per Member. As a fallback procedure, prior to storing the decision result in the Blockchain, the President can also request a new vote from all Members, thus generating a new decision.

The decision of each topic of a meeting consists of the result of a keyed hash function (HMAC) over a list of all votes, plus a session key. Using this approach, all decisions on all topics of all meetings will result in a different hash value because a new session key is generated per meeting. Otherwise, if a regular secure hash function was used, then there could be the possibility of the hash values of multiple decisions colliding with each other, for instance, in situations of multiple unanimous decisions.

## 5. Implemented Prototype

The prototype implemented to validate the proposed solution is depicted in Figure 6. The prototype comprises three distinct pieces of equipment: (1) a bACEVS server; (2) an authentication server; (3) a personal computer, simulating the President's workstation. The fourth piece of equipment, not shown, was also used and consisted of a smartphone, simulating a voting member. Both servers were set up as Virtual Machines (VM) on a host machine with 16 GB of RAM, an Intel I5 processor and 240GB SSD, running the MS Windows 10 operating system. Oracle Virtual Box version 6.0.14 was the adopted virtualisation platform. The bACEVS server VM was deployed with two cores and 8GB of RAM. The authentication server VM was deployed with two cores and 4GB of RAM. The President's workstation consists of a personal computer with 16GB of RAM, an Intel I5 processor and a 240GB SSD, running the MS Windows 10 operating system. The member's smartphone is an iPhone 6s with 4GB of RAM, and an Apple A9 processor, running iOS version 15.2.1.

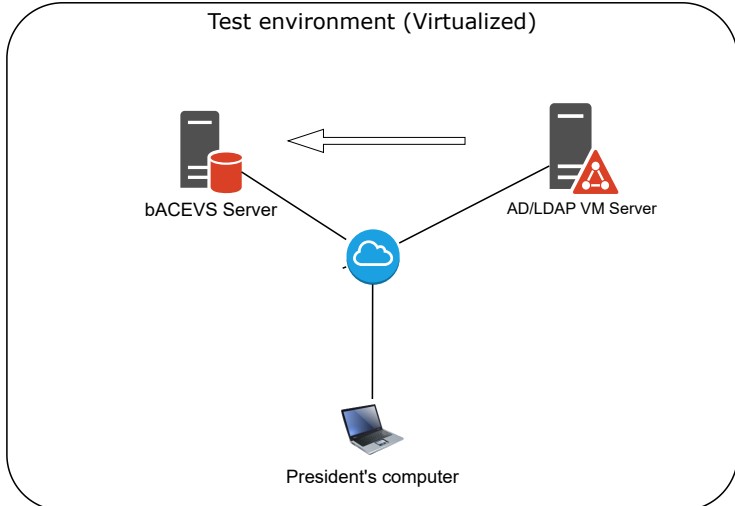

**Figure 6.** Test environment

*5.1. Software and Environment*

The software and tools that were used to implement the prototype are as follows. The bACEVS system consists of a web application built with PHP (version 7.3.31), running in an Apache web server (LAMP stack version 7.9.2009) deployed in a CENTOS virtual

machine. Versions CENTOS (version 7). The database used within the bACEVS was implemented using MySQL Workbench (version 8.0.29 CE). The authentication server consists of a Microsoft Active Directory (AD) server deployed in a Windows Server 2019 virtual machine. The blockchain was implemented with the use of IBM Blockchain Platform and its Local Test network. This platform implements the Hyperledger Fabric blockchain in a local network and can be integrated with the Visual Studio Code developer environment. In particular, IBM Blockchain Platform Extension for VS Code was used.

*5.2. Blockchain Interoperation*

The selected blockchain technology was the Hyperledger Fabric. The key reasoning behind this decision was based on applying the framework proposed in [50] to the reference scenario, resulting in the decision to adopt a DLT that imposes permissions on both read and write accesses but without requiring properties such as transparency or equal rights. On one hand, each peer or node in the system should store data from all councils of all faculties of the academic institution, to support redundancy and fault tolerance. On the other, each academic council should only be able to read information related to its own activity. The Hyperledger Fabric channel concept allows for this operation to happen naturally.

The Hyperledger Fabric world state can assume the form of a LevelDB (the default option) or of a CouchDB. LevelDB is a simple key/value storage, whereas CouchDB is a document storage allowing more complex queries to be performed. The CouchDB was selected. A copy of the ledger is maintained by each peer, including the blockchain, the chaincode and the world state.

To implement the interface between the Hyperledger Fabric and the bACEVS server, the relevant smart contract was implemented using IBM Blockchain Platform for VSCode version 2.0 and its Local Test network. The smart contract defines the executable logic that generates new facts to be added to the ledger. A smart contract is responsible for **put**, **get** and **delete** operations, which change the world state, and for making queries to the immutable ledger of transactions. A **put** creates a new business object or modifies it, a **get** queries the ledger to retrieve information about an existing business object and **delete** removes a business object from the current state of the ledger, maintaining its history. Generally, smart contracts decide what information is stored in the world state. Listing 2 shows the data structure used to store the decision tokens created per each topic voting act. The data structure comprises both the meeting topic identification (TopicID) and the HMAC of the decision record of the topic (DecisionRecordHMAC).

**Listing 2**: Decision data structure.

```
type Decision struct {
    topicid string 'json:"TopicID"'
    hashdecisonfile string 'json:"HashDecisionFile"'
}
```

A proof of concept smart contract was developed and deployed in order to demonstrate the feasibility of the proposed solution. The smart contract was named "DecisionContract", is shown in Listing 3 and comprises an initialization function named initLedger(), which creates some sample, hard-coded, decision tokens.

**Listing 3**: bACEVS smart contract for Decisions.

```
const { Contract } = require('fabric-contract-api');
class DecisionContract extends Contract {
async initLedger(ctx) {
    const decisions = [{
            topicid: 'T151120211435a9ab7fc521fd',
            hashdecisionfile: '477698a6-2f31-11ec-8d3d-0242ac130003',
        },{ topicid: 'T1511202114357f505aad471f',
            hashdecisionfile: '36989b61-6906-438d-8d32-15942-fee92b0',
        },{ topicid: 'T1511202116584903d950588f',
            hashdecisionfile: 'Tobdcb984e-7fe8-4be5-bc19-79ee97368331',
```

```
                },{ topicid: 'T1711302114367d505bbd417a',
                    hashdecisionfile: '088_f0fd4-e8c5-4a0e-b042-2517b8e95971',
                },];
            for (let i = 0; i < decisions.length; i++) {
                decisions[i].docType = 'decision';
                await ctx.stub.putState('DECISION' + i, Buffer.from(JSON.
                    stringify(decisions[i])));
            }
        }
    async decisionExists(ctx, decisionId) {
            const buffer = await ctx.stub.getState(decisionId);
            return (!!buffer && buffer.length > 0);
    }
    async createDecision(ctx, decisionId, value) {
            const exists = await this.decisionExists(ctx, decisionId);
            if (exists) {
                throw new Error('The decision ${decisionId} already exists');
            }
            const asset = { value };
            const buffer = Buffer.from(JSON.stringify(asset));
            await ctx.stub.putState(decisionId, buffer);
    }
    async readDecision(ctx, decisionId) {
            const exists = await this.decisionExists(ctx, decisionId);
            if (!exists) {
                throw new Error('The decision ${decisionId} does not exist');
            }
            const buffer = await ctx.stub.getState(decisionId);
            const asset = JSON.parse(buffer.toString());
            return asset;
    }
    async updateDecision(ctx, decisionId, newValue) {
            const exists = await this.decisionExists(ctx, decisionId);
            if (!exists) {
                throw new Error('The decision ${decisionId} does not exist');
            }
            const asset = { value: newValue };
            const buffer = Buffer.from(JSON.stringify(asset));
            await ctx.stub.putState(decisionId, buffer);
    }
    async deleteDecision(ctx, decisionId) {
            const exists = await this.decisionExists(ctx, decisionId);
            if (!exists) {
                throw new Error('The decision ${decisionId} does not exist');
            }
            await ctx.stub.deleteState(decisionId);
    }
}
module.exports = DecisionContract;
```

Additionally, five more functions were created, namely: decisionExists(), createDecision(), readDecision(), updateDecision(), and deleteDecision(). The decisionExists() function can be used to check for the presence of a specific decision token, using its identification. The createDecision() function can be used to add a new decision token to the ledger, if it is not already present on the ledger. The readDecision() function can be used to obtain the value of a decision token, using its identification. The updateDecision() function can be used to update the value of a decision token, using its identification. The deleteDecision() function can be used to delete a decision token, using its identification.

The next step is deploying the smart contract into a local test environment and, if it succeeds, testing whether transactions can be made invoking the previously described functions of the smart contract. Figure 7 shows the output information about a successful deployment of the smart contract.

```
[15/11/2021 20:12:28] [INFO] Open Transaction View
[15/11/2021 21:43:09] [INFO] Deploy Smart Contract
[15/11/2021 21:43:09] [INFO] connecting to fabric environment
[15/11/2021 21:43:10] [SUCCESS] Connected to 1 Org Local Fabric
[15/11/2021 21:43:10] [INFO] installSmartContract
[15/11/2021 21:43:11] [SUCCESS] Successfully installed on peer Org1 Peer
[15/11/2021 21:43:11] [INFO] approveSmartContract
[15/11/2021 21:43:13] [SUCCESS] Successfully approved smart contract definition
```

**Figure 7.** Output of local smart contract deployment.

## 6. Validation and Discussion

The list of requirements that were identified for the proposed solution can be found in Section 4. The first requirement (R1) states that the proposed solution must support multiple academic councils, which belong to multiple faculties, within one institution. To achieve this requirement, firstly the prototype implementation of the proposed solution integrates with an external authentication server. This way, the bACEVS does not need to maintain user profiles or passwords, increasing its overall security. Each time a user logs in, a validation request is made to the AD server. Figure 8 shows the implemented login form that enables this operation.

Authentication support was added to the prototype through the "login.php" file, shown in Listing 4. The process starts by validating if we are dealing with POST message that contains a username and password collected by the login form and then a connection to the indicated AD server (192.168.51.5) is made. In the case of a successful bind, a query to the users' database is made, filtered by "SAMAccountName" and an array with specific fields is filled (CN, fullName, mail, and displayName) to be used by bACEVS. If the bind is successful, it means the user exists in the authentication server and that the password is correct, otherwise, an "Invalid username or password!" message is displayed.

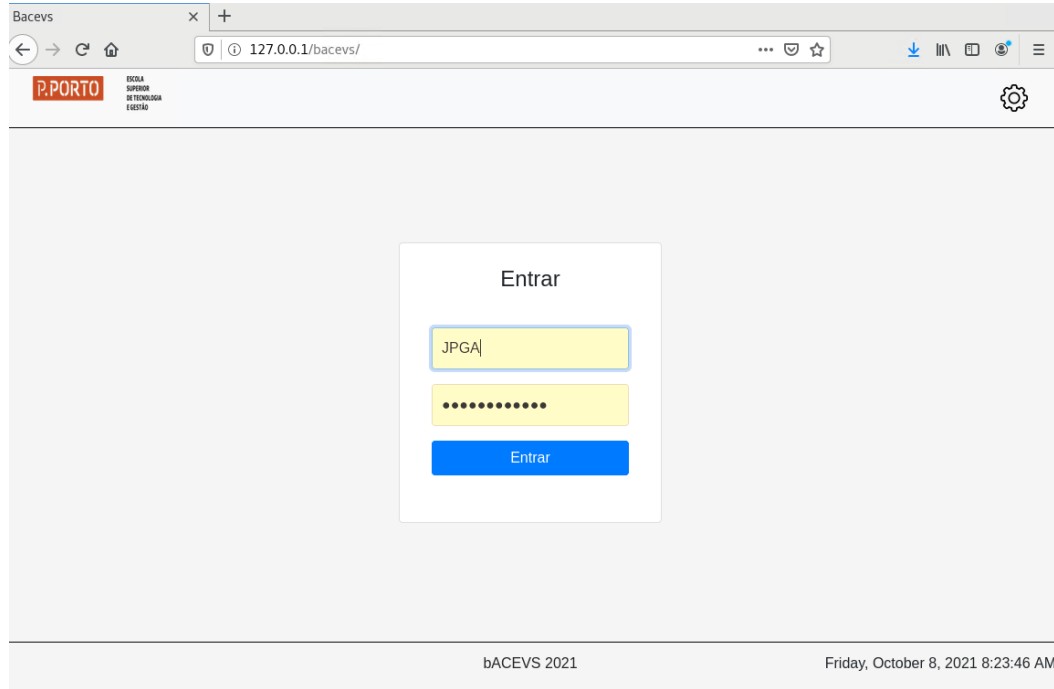

**Figure 8.** Login form of the bACEVS.



**Listing 4**: Login validation.

```php
if(isset($_POST['submit'])){
    $username=trim($_POST['userN']);
    $password=trim($_POST['pass']);
    $ldap = ldap_connect("192.168.91.5")
    or die("Could_not_connect_to_LDAP_server.");
    ldap_set_option($ldap, LDAP_OPT_PROTOCOL_VERSION, 3);
    $bind = ldap_bind($ldap, 'JPGADOMAIN\\'.$username, $password);
    if ($bind) {
        $dn = "DC=JPGADOMAIN,DC=LOCAL";
        $filter="SAMAccountName=$username";
        $justthese = array("CN","fullName", "mail","displayName");
        $sr=ldap_search($ldap,$dn, $filter,$justthese);
        $info = ldap_get_entries($ldap, $sr);
        $_SESSION['username'] = $info[0]['displayname'][0];
        header('location:'.$path.'/home.php');
    } else {
        $message = "Invalid_username_or_password!\\nPlease_try_Again.";
        header('location:'.$path.'/index.php?msg='.$message);
    }
}
```

Moreover, by opting for the Hyperledger Fabric as the adopted Blockchain, Requirement R1 can also be partially addressed. Hyperledger Fabric supports segmented use by being a permissioned Blockchain and allowing for the creation of channels, each channel having its own participants. Thus, one can set up a multi-node Hyperledger Fabric infrastructure. One node per school stores the information of all councils but still separates information access using segregation by channels.

Requirement R2 imposes that the bACEVS must adopt an operation based on the fact that members are elected for a mandate (period of time). The implemented prototype allows for the creation of mandates with a start and end date. After creating a mandate, it will be available for update allowing for the association of users to be part of this mandate. Figure 9 exemplifies such an operation. Marked with 1 and 2 (in green), we can see the start and end date associated with a mandate. Marked with 3, we can see an action button that opens a mini modal form (marked with 4) that allows for adding users from the database and selecting the role for each one.

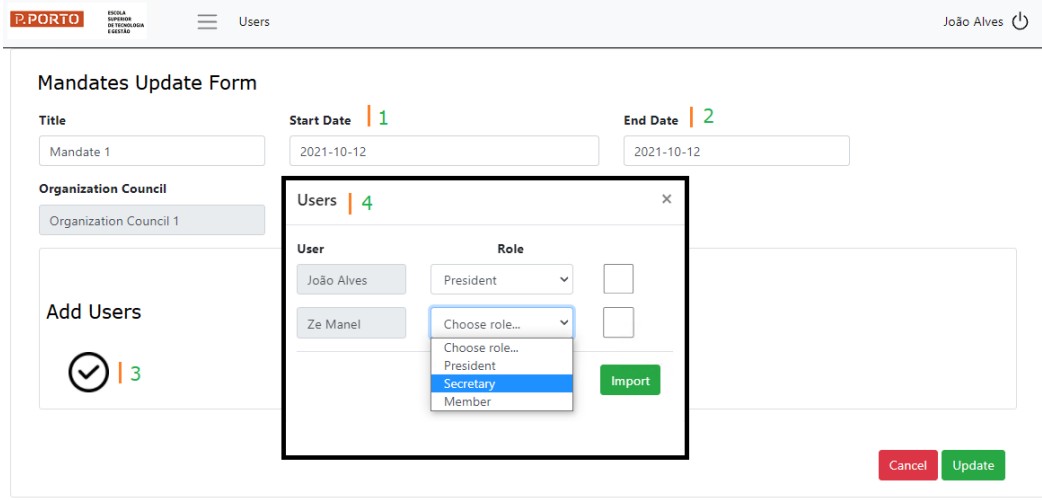

**Figure 9.** Mandate update form.

To address Requirement R3, CRUD operations were implemented in the prototype for all tables referred in Figure 10, marked with numbers 1 and 2 (in green). Administrators can add records, view, update and delete them using the "Tables" menu entry. Requirement R4 is related to the scheduling of meetings. The prototype fulfils this feature by asking the

user to create a meeting on a specific date using a calendar-based form. For convenience, the prototype also shows the calendar as its background on the main menu (see Figure 10). All scheduled meetings are visible in the calendar.

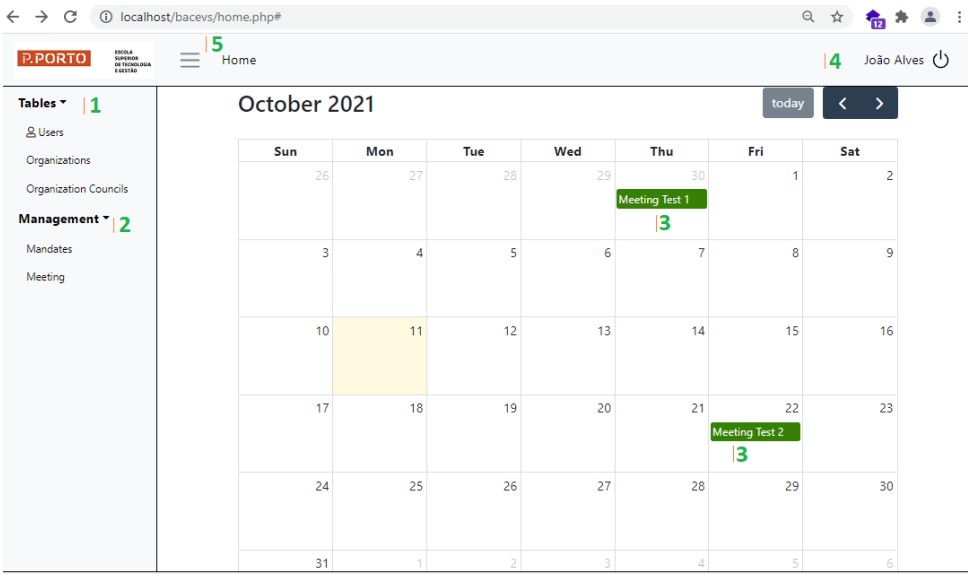

**Figure 10.** Main menu.

Requirement R5 addresses topic management within meetings. The implemented prototype allows for topics to be inserted into an existing meeting, in the order that these are to be discussed. The "Meeting" form of the prototype shows the general information about the selected meeting and a blue button, in the form of a plus symbol. This button can be used to add topics. Figure 11 shows multiple topics added to an existing meeting.

**Figure 11.** Meeting update form

To achieve Requirement R6 of associating external documents, in the pursuit of the full digitalization of the decision-making process, the prototype enables per-topic document uploading. These documents will also be stored in the database, for archive and historic purposes. Figure 12 illustrates the addition of a new topic with a file attached.

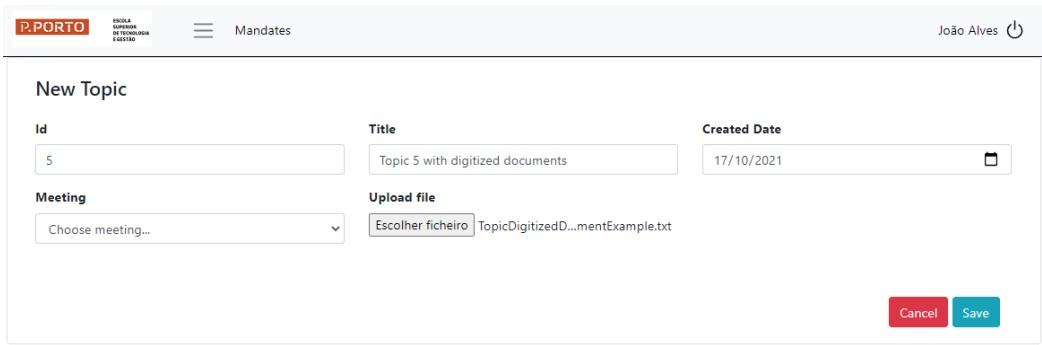

**Figure 12.** New Topic form

Requirement R7 imposes that the proposed solution must record member participation in meetings, including their vote and the overall decision that was made, based on the vote count. This requirement was implemented by adding a "presence" field to the table in the database that stores the meeting details. This field is automatically filled by the prototype whenever a member enters an open meeting. Votes are also registered in a specific table within the database, allowing the prototype to know the position taken by each member on each topic, as well as being able to count the votes and compute a final decision on each topic.

Requirement R8 imposes that the proposed solution must guarantee the immutability of records of the voting acts and, in the implemented prototype, was performed by using an Hyperledger Fabric ledger to store such information. Additional details regarding this integration were previously described in Section 5.2.

As a summary of the previously validated requirements, bACEVS supports an operation based on the fact that members are elected for a mandate in a period of time. It allows for the creation of records for mandates that have well-defined start and end dates. Supports multiple academic councils, in multiple faculties within one institution. In the management of the academic councils, users can be added from the "User" table, assigning a role to each one, such as president, secretary or member. Meeting management is also possible. Figure 10 shows the main menu of implemented prototype and its main components.

Managing meetings is simplified by interacting with the calendar object to select the date of a scheduled meeting. A meeting can be opened, triggering an email notification to be sent to all participating members. The email notification includes a list of all the topics available for discussion in the meeting. Figure 13 illustrates the options made available to the President in regard to meeting management, after being opened. Options are the closing of the meeting or the requesting of votes by members on each topic.

At this point, members accessing the prototype using their mobile browser application can interact with the meeting sessions by voting on topics whenever these topics are opened to vote by the President. Figure 14 illustrates the mobile version of the prototype interface that enables members to vote on a topic.

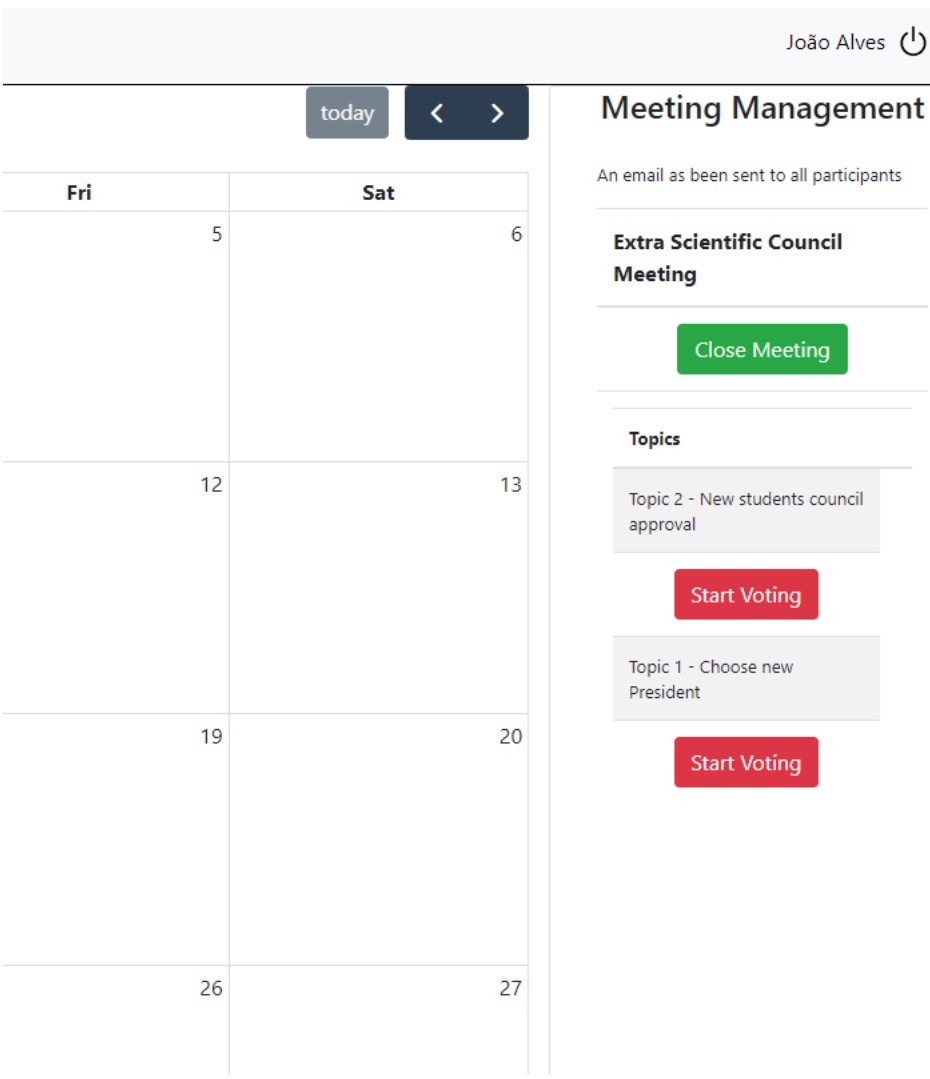

**Figure 13.** Managing meetings.

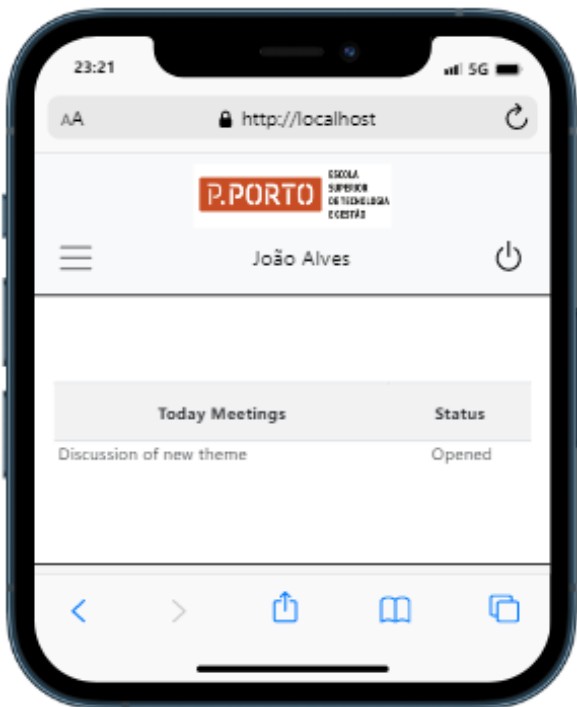

**Figure 14.** Mobile application.

## 7. Conclusions

The COVID pandemic forced remote work upon universities and polytechnic institutes and their academic councils. Academic councils take decisions based on nominal voting, without anonymity. A blockchain-assisted voting system can fully support the decision-making process of academic councils in remote, online meetings. The proposed bACEVS system implements such an approach while offering additional guarantees such as vote integrity and verification. The adopted blockchain allows for having a node per faculty in the university, supporting redundancy but still enforcing council confidentiality. A prototype of the proposed solution was implemented, tested and validated.

Smart cities are expected to become a more present reality in the future, where smart environments make use of technology to promote a better way of life. In such a technological environment, electronic voting solutions such as the one proposed herein are expected to exist. In the future, the authors will validate the applicability of the bACEVS to other processes of group decision-making in smart cities. Examples are city councils and high or elementary school councils, where new and different security and privacy requirements may appear. A detailed performance evaluation is also expected to be pursued.

**Author Contributions:** Conceptualization, J.A. and A.P.; software, J.A.; validation, J.A. and A.P.; investigation J.A.; writing—original draft preparation, J.A. and A.P.; writing—review and editing, J.A. and A.P.; supervision, A.P. All authors have read and agreed to the published versions of the manuscript.

**Funding:** This research received no external funding.

**Institutional Review Board Statement:** Not applicable.

**Informed Consent Statement:** Not applicable.

**Data Availability Statement:** Data sharing not applicable.

**Conflicts of Interest:** The authors declare no conflict of interest.

## Abbreviations

The following abbreviations are used in this manuscript:

| | |
|---|---|
| AD | Active Directory |
| bACEVS | Blockchain.Assisted Academic Council |
| BFT | Byzantine Fault-Tolerant |
| CA | Certificate Authority |
| CENTOS | Community ENTerprise OS |
| CN | Common Name |
| DApps | Distributed Applications |
| DEX | Decentralized Exchanges |
| DLT | Distributed Ledger Technology |
| ERC-20 | Ethereum Request for Comments - Number 20 |
| EVS | Electronic Voting System |
| GB | GigaByte |
| ICO | Initial Coin Offerings |
| JSON | JavaScript Object Notation |
| LAMP | Linux, APACHE, MySQL and PHP |
| LDAP | Lightweight Directory Access Protocol |
| PoW | Proof of Work |
| RAM | Random Access Memory |
| SSD | Solid State Drive |
| SEK | Session Encryption Key |
| TLS | Transport Layer Security |
| VM | Virtual Machine |

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
