# Peer review of "On the Implementation of a Blockchain-Assisted Academic Council Electronic Vote System"

_smartcities, doi:10.3390/smartcities6010014_

Round 1

Reviewer 1 Report

1. A separate subsection like “Software and Environment” to present the used Software, tools, environment configuration , packages and utilities under section 5 would benefit the readers.

2. Optionally in Results and discussion if you could compare or present the performance metric of your implemented system would be great. (optional)

3. A data flow diagram to represent the data transition from the front end (web / mobile) to the blockchain would help future researcher and developer to understand better and refer your work.

4. Please defend – The storing of data in your local SQL database (line 442) and again in the blockchain (line 445) would lead to data redundancy and might affect the convergence as multiple nodes may lead to isolated data islands.

5. In which place / step of the process is the HMAC used and please justify the need. Because, Ultimately the transaction gets hashed in the blockchain.

6. In future works the authors may also consider improving the security and privacy aspect of the system.  The work requires some defence to the design choice and overall the paper is a needed innovation for this modern age. 

I appreciate the author and would like to see an real-world implementation of this system for academia in future.

Reviewer 2 Report

This paper proposes a decision-making method, by voting in Academic Councils, to be supported by a system that guarantees the integrity of the decisions taken, even when meeting online.

1. The performance evaluation of the proposed method is too simple, and it lacks the comparison with other advanced systems proposed in recent three years.

2. Many symbols, notations are not well designed, and should be simplified. A massive number of symbols are introduced in the formation. It will help understanding the optimization by providing a proper nomenclature at the beginning of this paper.

3. The novelty of this paper needs to be strengthened. The authors need to add more details in contribution bullet points that will clearly show readers the main contribution of this work.

Reviewer 3 Report

Firstly, I would like to congratulate the authors for their effort in writing the article On the Implementation of a Blockchain-assisted Academic Council Electronic Vote System. E-voting, not to mention blockchain technologies, are rather new topics, interesting and very much debated in the actual researches.  

However, before publishing, few details need to be addressed:

1.       In Figure 1 (also in the explanations below it) one can see that the voter (council member) is sending the vote decision to the platform directly and not by a the blockchain system. If so, how are the votes going to be anonymized? I understand that this is a working model developed by others but using it as a reference here (considering vulnerable) it makes little sense.

2.       In respect to other authors when making statements like on line 65, I suggest naming them together with the year and add the reference as it is now afterwards. It’s not academic to just say: The term [whatever] originally appeared in [6] and consisted [...] even if 6 is a reference. Kindly adjust as on line 132 or 264 (funny coincidence in the line numbers).

3.       Also rephrase line 75… some comment as before.

4.       Line 81… is it always rewarded with bitcoins? I don’t see the context here… please clarify.

5.       On figure 2, the block 1 is using the hash of previous block header. It seems difficult since is the first one. Trying to search for the reference it was also difficult since not too much info about it is provided. I found this however, https://www.ussc.gov/sites/default/files/pdf/training/annual-national-training-seminar/2018/Emerging_Tech_Bitcoin_Crypto.pdf... but not that much info in it that actually support figure2. My suggestion here is to re-adapt the figure and also update the reference (I will come back to that later on this review).

6.       Line 140. No reason to add “in” before [16] and no reason to use ().

7.       Line 188… kindly add reference at the end of the cited paragraph or phrase (same in the case of comment 3 here with regard to reference [6] and [7]).

8.       On line 194 there is the same approach as the one mentioned already in the comment 2 here. Please make sure around the whole article not to repeat this… I won’t make any new observation on this but I consider the issue as very important. Please use the same approach as it was used on line 132 / 264. It shows respect and consideration for fellow researchers and their work.

9.       Some statements, such as in the paragraph between line 232 and 241 (but not only, the very next paragraph has the same problem), needs references. Intuitively, the arguments are plausible, but we are not here to express our intuition, but to prove it. Please consider adding references.

10.   Line 242 is almost similar with 233… please rephrase.

11.   Kindly make figure 3 more coherent…

12.   While reading the article, although a very good description of the technology, I think it is missing something and that is how the system can protect itself against multiple voting by the same person. So, we have M and an IDm for any given meeting, topic and so on… What if before taking the vote, but of course after an IDm is generated, so after he gets accepted as a member, he is losing temporarily the availability of service? We have 2 scenarios here: (1) If another IDm will be generated, how can we be sure that he lost the availability (e.g., connection) before voting? (2) he loses the right to vote (the system won’t generate another IDm). Both scenarios are undesirable. I tried to understand if this issue is already solved by the diagram in fig 5, but I couldn’t see. Please explain in the article how you secure this or, if is already explained, please point it more clearly… maybe I missed it.

13.   The conclusion section seems to be more like a results section. I suggest renaming it as such. I would like seeing a further application of the model section as well. Here the authors might try connecting the already addressed topic with the Smart Cities concept… yet I see little to no connection between the two. Maybe, as a suggestion if I may, proposing this also to the city councils or proposing the model to schools or high schools in cities on their aim to become smart and so on.

14.   In regard with the references. Most of them (not all of them though - e.g., [6] it has nothing else then the title and author) are in good shape; however, I would like to make a strong suggestion to the authors and that is to find inspiration on the MDPI Smart Cities journal, there are plenty of great article that could be used to draw this research in an even brighter way. Also try looking at the SCRD Journal (scrd.eu), also some good articles to be found there. After searching the mentioned journals (and of course others) my previous suggestion here (no. 13) could be solved almost by default.

Overall, the article worth being published after following the above suggestions. I congratulate the authors for their initiative and efforts.

Round 2

Reviewer 1 Report

Please clarify line 458 "The nonce numbers will be used to identify the meeting actions.". If correction required please correct this before final acceptance. 

Line 529 - what are the remaining data?

Line 535 - why 3 nodes ? Local copies will always lead to multiple challenges in blockchain rather than advantages stated in line 533.

In future work or for security , privacy or high performance signature concepts research work like https://doi.org/10.1016/j.jksuci.2021.12.001 can be directly cited and referenced.

For metrics and future improvement article like 10.1109/ACCESS.2020.3013282 can be referred.
